# Year-Round Presence of Microcystins and Toxin-Producing *Microcystis* in the Water Column and Ice Cover of a Eutrophic Lake Located in the Continuous Permafrost Zone (Yakutia, Russia)

**DOI:** 10.3390/toxins15070467

**Published:** 2023-07-20

**Authors:** Viktor A. Gabyshev, Sergey I. Sidelev, Ekaterina N. Chernova, Anna A. Vilnet, Denis A. Davydov, Sophia Barinova, Olga I. Gabysheva, Zoya A. Zhakovskaya, Ivan V. Voronov

**Affiliations:** 1Institute for Biological Problems of Cryolithozone, Siberian Branch, Russian Academy of Sciences, Yakutsk 677980, Russia; v.a.gabyshev@yandex.ru (V.A.G.); g89248693006@yandex.ru (O.I.G.); viv_2002@mail.ru (I.V.V.); 2Faculty of Biology and Ecology, Yaroslavl State University, Yaroslavl 150057, Russia; sidelev@mail.ru; 3Papanin Institute for Biology of Inland Waters, Russian Academy of Sciences, Borok, Yaroslavl 152742, Russia; 4Scientific Research Centre for Ecological Safety, St. Petersburg Federal Research Center, Russian Academy of Sciences, St. Petersburg 197110, Russia; s3561389@yandex.ru (E.N.C.); zoya.zhakovskaya@gmail.com (Z.A.Z.); 5Polar-Alpine Botanic Garden-Institute—Subdivision of the Federal Research Centre “Kola Science Centre”, Apatity 184209, Russia; a.vilnet@ksc.ru (A.A.V.); d.davydov@ksc.ru (D.A.D.); 6Institute of North Industrial Ecology Problems—Subdivision of the Federal Research Center “Kola Science Center”, Apatity 184209, Russia; 7Institute of Evolution, University of Haifa, Mount Carmel, 199 Abba Khoushi Ave., Haifa 3498838, Israel

**Keywords:** year-round study, ice, cyanobacteria, *Microcystis* spp., microcystins, continuous permafrost zone, microcystin quota

## Abstract

This study aimed to test the hypothesis of the year-round presence of toxigenic *Microcystis* and cyanotoxins in the water and ice of the shallow eutrophic Lake Ytyk-Kyuyol located in the continuous permafrost zone. Three independent approaches—mass-spectrometry, molecular methods and light microscopy—were applied in the study. The cyanobacterial biomass ranged from 1.0 × 10^−4^ to 4.8 mg L^−1^. *Microcystis flos-aquae* and *M. aeruginosa* were the dominant morphospecies in plankton throughout the observation. In environmental DNA, the presence of *M. aeruginosa* was supported and *mcy* gene regions responsible for microcystin biosynthesis were detected through a BLAST (Basic Local Alignment Search Tool) search and phylogenetic estimation based on newly obtained 16S rRNA, 16S–23S ITS rRNA, *mcy*A and *mcy*E nucleotide sequences. The intracellular microcystin concentration ranged from <0.1 to 803 ng L^−1^, and the microcystin quota in the *Microcystis* biomass was extremely low. For the first time, it was shown that *Microcystis* cells containing *mcy* genes and microcystins presented permanently in the water column, both during the ice-free period and under ice, as well as inside thick ice covers within 7 months of severe winter. We hypothesized that minor pelagic and ice populations of *Microcystis* could participate in increasing cell density in the spring. However, further studies are needed to confirm the viability of the overwintering *Microcystis* colonies in the water and inside the ice of Lake Ytyk-Kyuyol.

## 1. Introduction

Cyanobacteria colonize a wide range of habitats and can occur under very severe temperature conditions [1,2,3,4]. One of the harmful features of cyanobacteria is their ability to produce a broad panoply of toxic compounds, among which the most common are hepatotoxic microcystins (MCs), first isolated from the cyanobacterium *Microcystis aeruginosa* [5]. Most research is devoted to the distribution of cyanobacteria and their toxins in temperate and tropical water bodies, where they are able to form cyanobacterial harmful algae blooms (Cyano-HAB), which are promoted, along with increased eutrophication by warm climates, and the optimal water temperature for the growth of cyanobacteria [6,7,8,9]. The growing interest in cyanobacterial persistence at relatively low temperatures has been noted in recent years. Changes in climatic parameters may affect the structure of algal communities of reservoirs, with a shift in favor of toxin-producing species of cyanobacteria in the permafrost zone [10]. Not surprisingly, they have been studied by algologists for a very long time.

A cosmopolitan distribution characterizes the *Microcystis* species, but in the high-latitude regions of the northern hemisphere, the number of their records is small. The northernmost known localities of *Microcystis* spp. are occurrences on Ellesmere Island in the Canadian Arctic Archipelago [11], Franz Josef Land [12] and Taymyr peninsula [13]. Recent works reflect the presence of cyanotoxins in reservoirs of the northern and polar regions [14,15,16,17], where MCs were detected in the plankton of 18 Greenland lakes [18] and in the subarctic Lake Imandra of the Kola Peninsula [16].

In an increasing number of studies, in winter, ice blooms of cyanobacteria often dominated by toxic *Microcysts* have been reported in temperate reservoirs [19,20]. *Microcystis* is known to be an MC-producing colony-forming cyanobacterium, with a cosmopolitan distribution throughout the world. It was noted that in temperate lakes, *Microcystis* settled into the sediment during the winter, could be reintroduced into the water column in the spring, and rise to the water surface where it could form blooms [5]. It has also been assumed that, in addition to the bottom, *Microcystis* are able to winter as a pelagic population in the subglacial water column in shallow eutrophic lakes [19,21]. It could be supposed that ice is another habitat in the reservoir, where *Microcystis* colonies containing MC persist in winter. We were able to find only one report of a natural toxigenic population of *Microcystis* of which the cells remained viable after freezing into ice in a small pond in Hungary [22].

Recently, we discovered the presence of a *Microcystis* spp. in the summer plankton of some lakes in the permafrost zone in Yakutia, Russia. A continuous permafrost zone is a very special climatic zone in which permafrost occurs everywhere beneath the land surface. The climate is severely continental, with harsh, long winters and short, hot summers. The coldest inhabited place on Earth—Pole of Cold, Oymyakon—is in the Yakutia region. Among the studied lakes in this region, signs of cyanobacterial bloom were visually detected in Lake Ytyk-Kyuyol [23]. These initial data prompted us to conduct a study on the year-round presence of toxigenic cyanobacterial species in the lake.

This study aimed to test the hypothesis of the year-round presence of toxigenic *Microcystis* and cyanotoxins in water and ice of the shallow eutrophic Lake Ytyk-Kyuyol located in the continuous permafrost zone, and estimate the toxigenicity level of the *Microcystis* biomass. From the entire set of ultrasensitive methods for studying toxigenic cyanobacteria [24,25,26,27,28], we used three independent approaches to achieve this aim— high-performance liquid chromatography–high-resolution mass-spectrometry (HPLC–HRMS), molecular methods and light microscopy—which allow the reliable identification of cyanotoxin producers in water bodies, and the quantification of cyanotoxins in various matrices (water, ice, plankton).

## 2. Results

### 2.1. Physico-Chemical Parameters

The water temperature in the open water period ranged from 4.6 °С (September 2022) to 27.1 °С (July 2022). The air temperature increased to 35 °C in the summer and dropped to minimal values of −35–−42 °C over the winter. The temperatures of the surface layers of under-ice water were extremely low (−0.1–−0.8 °С) (Figure 1). The bottom water layer was cooler by 0.7–3.5 °C compared to the surface water layer during the open water period and, vice versa, i.e., a warmer by 1.0–2.4 °C during the ice cover period. The ice cover thickness increased during the winter, reaching its maximum of 120 cm in April (Figure 1).

The water pH was slightly alkaline (Table 1). Maximum values of salinity and hardness of water were recorded at the end of the ice cover period. The water color index varied over a wide range, reaching maximum values in the middle of the ice-free period. The content of total phosphorus and mineral phosphorus, ammonium nitrogen, nitrate nitrogen and nitrite nitrogen was high over the entire study period, with a maximum during the ice cover period (Table 1). Waters were characterized by elevated chemical oxygen demand (COD) values. Thus, Lake Ytyk-Kyuyol was characterized as a highly eutrophic water body undergoing a significant nutrient load and organic pollution.

### 2.2. Composition and Seasonal Dynamics of Cyanobacterial Community

In our study period of August 2021—November 2022, in the phytoplankton of Lake Ytyk-Kyuyol, 33 species of cyanobacteria were identified (Appendix A). The identified morphospecies belonged to 16 genera; among them, *Microcystis* (seven species), *Aphanocapsa* and *Dolichospermum* (four each), *Anabaena* (three), *Anathece*, *Merismopedia* and *Oscillatoria* (two each) were the most diverse. *Aphanizomenon flos-aquae*, *Aphanothece microscopica*, *Coelomoron pusillum*, *Gomphosphaeria aponina*, *Kamptonema chlorinum*, *Phormidium ambiguum*, *Rhabdogloea smithii*, *Snowella lacustris* and *Woronichinia naegeliana* were the only ones representative of each genus. The most diverse cyanobacterial community has been noted during the open water period. The number of species varied from 9 to 12 (Appendix A). The cyanobacteria species richness of under-ice plankton was statistically significant (Mann–Whitney test, U = 0, *p* < 0.001), and dramatically decreased to 4–5 species per sample; by the end of the period of ice phenomena, only one species (*Microcystis flos-aquae*) was found in the under-ice plankton sample collected in April.

The number of species of planktonic cyanobacteria frozen in ice ranged from 3 to 7. *Aphanocapsa grevillei*, *Aphanothece microscopica*, *Coelomoron pusillum*, *Microcystis aeruginosa*, *M. botrys*, *M. flos-aquae*, *M. wesenbergii*, *Rhabdogloea smithii*, *Snowella lacustris* and *Woronichinia naegeliana* were discovered in ice samples*. M. flos-aquae* was the species found in all examined ice samples (Figure 2, Appendix A).

Seasonal dynamics of the abundance and biomass of planktonic cyanobacteria in Lake Ytyk-Kyuyol were characterized by one peak, which was noted in August (Figure 3). The maximal vegetation level of cyanobacteria, with an abundance of 342.6 × 10^6^ cells L^−1^ and a biomass of 4.8 mg L^−1^, was reached in August 2021. The contribution of five potentially toxigenic representatives of the genus *Microcystis* (*M. aeruginosa*, *M. flos-aquae*, *M. wesenbergii*, *M. botrys* and *M. novacekii*) accounted for a significant part of the cyanobacterial plankton throughout the entire observation period, ranging from 23 to 100% of the total abundance, and 39–100% of the total biomass.

In the first month of the ice cover period (October), under-ice plankton still retained the vegetation level that was noted at the end of the open water period. A sharp decrease in cell concentration and biomass of cyanobacteria, including *Microcystis*, occurred in November and decreased to the minimum in under-ice water samples in April (Figure 3). A rapid increase in the cell density and biomass of planktonic cyanobacteria began in May, immediately after the destruction of the ice cover, with the onset of water warming and a significant increase in daylight hours (Figure 3).

The maximum abundance and biomass of the planktonic community of cyanobacteria frozen into ice were observed at the beginning of the period of ice events, in October. Thus, in the ice sample 6 cm thick collected in October 2022, the maximal abundance of *M. aeruginosa*, *M. flos-aquae*, *M. botrys*, *M. wesenbergii* and *A. grevillei* was found. The abundance of cyanobacteria gradually decreased in ice samples at the end of the ice cover period in April (Figure 3).

### 2.3. Molecular Identification of Uncultured Cyanobacteria

The tested environmental DNA sample revealed a single PCR (polymerase chain reaction) product of the 16S rRNA gene and the 16S–23S ITS rRNA sequence that was assembled and deposited in GenBank under accession number OR147468. We provided a BLAST search with type material only. We determined the similarity of the tested sample with the strain type of *Microcystis aeruginosa* NIES-843 with 98.61% in the whole region of the gene and spacer (AP009552) and 99.39% in 16S rRNA (NR074314). The maximum likelihood (ML) analysis of the 16S rRNA gene dataset resulted in a single tree with an arithmetic mean of log-likelihood −6396.322. The obtained tree topology is shown in Figure 4. The tested sample from Lake Ytyk-Kyuyol was placed in a clade composed of nine *Microcystis* strains that presented five morphospecies, including the strain type *Microcystis aeruginosa* NIES-843, with 98% bootstrap support. The *p*-distance estimation (Table 2) showed 98.56–99.86% 16S rRNA gene similarity between the tested sample and other *Microcystis* strains, and 99.64% similarity with strain type NIES-843 that lay within the infraspecific threshold (<1.3%) for cyanoprokaryotes [29]. Our results indicated that the tested strain was *Microcystis aeruginosa*.

### 2.4. Cyanotoxin-Producing Genes

Fragments of *mcy*A and *mcy*E genes were amplified from all plankton samples using general MC-producing cyanobacteria primers (*mcy*ACd and HEP). For the taxonomic identification of МС producers, specific МС-producing *Microcystis* primers (*mcy*A_MF/MR and *mcy*EF2/MicmcyE-R8) were used, and identical positive results were obtained. The ice samples were analyzed the same way. All collected ice samples had fragments of *mcyA* and *mcyE* genes specific to *Microcystis*. An example of their detection using a PCR method for the studied samples is shown in Figure 5.

The PCR products were sequenced to verify that the *mcyA* and *mcyE* genes were amplified. Nucleotide sequences were deposited into GenBank under accession numbers OQ971349 (*mcyE*, 458 bp) and OQ971350 (*mcyA*, 293 bp). The BLAST analysis of the amplified PCR products showed that the sequences were similar to *mcyA* and *mcyE* genes of *Microcystis* species (identity > 99.5%, coverage—99%). The phylogenetic analysis confirmed the BLAST results demonstrating that there were three distinct *Microcystis*, *Planktohtrix* and *Anabaena* (*Dolichospermum*) clades of *mcyA* and *mcyE* genes (Figure 6 and Figure 7). The partial sequences of *mcyA* and *mcyE* genes from uncultured *Microcystis aeruginosa* cyanobacterium inhabited in Lake Ytyk-Kyuyol were clustered with *Microcystis* sequences, with 100% bootstrap support (Figure 6 and Figure 7).

### 2.5. Cyanotoxins

Among all cyanotoxins, only MCs were detected. The МС profile was represented by common arginine-containing congeners. In total, eight congeners were identified: [D-Asp^3^]MC-LR, MC-LR, [D-Asp^3^]MC-RR, MC-RR, MC-YR, [D-Asp^3^]MC-YR, MC-WR, [D-Asp^3^ and ADMAdda^5^]MC-LR. The example of the extracted ion chromatogram of a high resolution (mass accuracy within 5 ppm) for MC congeners detected in the biomass filtered from surface water (sampling date 11 August 2022, volume 0.5 L) and ice (sampling date 24 October 2022, volume 1.0 L) is presented in Figure 8.

Throughout the year, only 2 out of 15 surface water samples (sampling dates 24 March and 22 April) did not contain intracellular MCs (detection limit, LOD 0.1 ng L^−1^) (Figure 9). Furthermore, none of the samples (water and ice) collected at these sampling dates contained detectable amounts of MCs. The maximal intracellular MC concentration (803 ng L^−1^) was detected in the surface water sampled in August 2021. Seven MC congeners were identified in the sample. Among them, the share of MC-LR and MC-RR with the total content was the highest. In the ice samples, the highest MC concentration (48 ng L^−1^) was detected in October, the first sampling event after ice covering. In the ice samples, up to five MC congeners were identified. Similar to water samples, two MC variants, namely MC-LR and MC-RR, made the greatest contribution. In the sample collected on October 24 2022, the share of MC-LR and MC-RR amounted to 56% and 30% of the total content, respectively (Appendix A). The extracellular MC concentration determined in the filtered water of some surface water samples was insignificant (Figure 9, Appendix A). The variations in total MC concentrations are shown in Figure 9. The calculated MC quotas per unit biomass of cyanobacteria producers (B) were in the range of 0.2 × 10^−2^ 0.31 μg MC mg^−1^ B.

## 3. Discussion

Despite the excess of nutrients, cyanobacterial bloom in Lake Ytyk-Kyuyol was limited to a short vegetation period (from the end of June to the end of August) due to the severe climatic features of the permafrost zone. The maximum levels of cyanobacteria were reached in early August, which was noted by researchers earlier. According to [30], the mass growth of cyanobacteria was noticed in the lake in July 1964, when their biomass reached 8.5 mg L^−1^. This study recorded maximal values of cyanobacterial biomass of 4.8 mg L^−1^ and 4.5 mg L^−1^ in August 2021 and 2022, respectively. Moreover, the shares of potentially toxigenic *Microcystis* spp. in these samples were 67% and 89% of the cyanobacterial biomass.

In the study, the year-round presence of toxigenic *Microcystis* in the shallow eutrophic lake under severe climate conditions of a continuous permafrost zone was demonstrated for the first time. Using three independent and complementary methods (light microscopy, molecular techniques and HPLC–HRMS), it was shown that *Microcystis* cells containing MC-producing genes and toxins were permanently present in the water column, both during the ice-free period and under the ice. Intracellular МСs were only not detected in the samples of under-ice water collected in March and April 2022, as their concentration decreased below the method’s detection limit. In general, the seasonal dynamics of *Microcystis* abundance and intracellular content of MCs related tightly was proved by a rather high correlation coefficient between these parameters (r_s_ = 0.88, *р* < 0.001, *n* = 25). Despite under-ice blooms, including *Microcystis* blooms, likely being a rather widespread phenomenon, especially in shallow eutrophic lakes [4,16,22], the abundance of the *Microcystis* population in Lake Ytyk-Kyuyol dramatically decreased in under-ice water compared to the ice-free period without *Microcystis* bloom-forming. *Microcystis* includes 65 taxonomically accepted species [31]. However, lineage separation for *Microcystis aeruginosa*, *M. flos-aquae*, *M. ichthyoblabe*, *M. novacekii*, *M. panniformis*, *M. protocystis*, *M. viridis* and *M. wesenbergii* based on 16S rRNA sequences was not possible [5]. In this study, we considered morphologically identified *Microcystis* species as morphospecies, with molecular support only for *M*. *aeruginosa*.

*Microcystis botrys*, *M. novacekii* and *М. wesenbergii* disappeared from phytoplankton in the under-ice water. However, *M. flos-aquae* and *M. aeruginosa* had an abundance not exceeding 1–10 thousand cells per liter in under-ice water at the extremely low temperature of 0.1–0.8 °С and air temperature −35–−42 °С. It is obvious that among the main limiting factors in the winter besides low water temperature, the lack of light due to both the thick ice cover (>1 m) and the short daylight hours is of great importance for the water bodies located in the 62nd parallel. It has been known that lake sediments are the main depositary for the over-wintering population of *Microcystis*, maintaining their viability [32,33,34,35]. *Microcystis* colonies due to buoyancy control can sink to the bottom where the temperature is higher compared to the surface, and settle into the sediments for overwintering (Figure 10). In spring, when the water column warms up and the penetrating solar radiation increases, the overwintering benthic population of *Microcystis* rises into the water and starts to grow. For water bodies of the temperate zone, it has been shown that a small or, in some cases, a significant amount of the overwintering population of *Microcystis* can persist in the pelagiс zone of lakes [19,21,36]. This part of the population seems to be of great importance, since it can determine the level of water bloom in the subsequent summer period [37]. It is still little understood how pelagic populations of *Microcystis* maintain their existence in adverse physical conditions under the ice, whether they are metabolically active, or whether their presence in the water column is the result of a constant resuspension of bottom sediments, where the overwintering benthic part of the population is located [4,19,35]. However, to our knowledge, our study is the first report on the ability of *Microcystis* to maintain a small pelagic population in a lake for most of the year under ice during the harsh winters of the continuous permafrost.

A third depositary may potentially exist in water bodies: ice cover, where *Microcystis* cells may remain dormant during the winter and can inoculate the water column in the spring (Figure 10). However, data on the findings of viable colonies of *Microcystis* in ice are scarce and limited to water bodies of the temperate zone. Thus, in the article [21], the authors discuss a case study in the pond Hármashegy (Hungary), where a large number of viable *M. viridis* cells containing MC-RR were found in the ice cover. However, the seasonal dynamics of *Microcystis* cell density and MC concentrations in ice have not been studied. In our study, for the first time, colonies of *M. flos-aquae* and *M. aeruginosa* with genes encoding MC production were found inside the ice cover of Lake Ytyk-Kyuyol. The presence of intracellular MCs was recorded in almost all the ice samples except for two collected at the end of the ice period in March and April, where the MC content fell below the detection limit of the HPLC–HRMS method. We assume that the *Microcystis* colonies frozen in ice and the MCs contained in cells should have been well-preserved during the ice period (Figure 10). However, we observed a dramatic decrease in the abundance of toxigenic *Microcystis* and the concentrations of intracellular MCs in the ice samples collected from December 2021 to April 2022, and October to November 2022 (Figure 3 and Figure 8, Appendix A). This phenomenon could be explained by the fact that during the winter with extremely low temperatures, the ice thickness in Lake Ytyk-Kyuyol grew due to the under-ice water, which practically did not contain *Microcystis* colonies by the middle of winter. Thus, the “effect of dilution” of ice occurred. Our assumption was confirmed by the fact that the abundance of *Microcystis* and the intracellular concentrations of MC decreased simultaneously from December 2021 to April 2022, and from October to November 2022 in the under-ice surface water layer (Figure 3 and Figure 8). A comparison of the appearance of *Microcystis* colonies frozen in ice at the beginning (November) and at the end of winter (April) with summer *Microcystis* colonies did not reveal any noticeable cell damage during ice thawing (Figure 2). Although our data cannot reliably confirm the viability of *Microcystis* frozen in the ice of Lake Ytyk-Kyuyol, some published studies indicate a high resistance of this genus to freezing and overwintering inside the ice. In the study [22], it was shown that of the total viable colonies of *M. viridis*, 58% appeared in the ice cover, 40% were identified in the sediment and only 2% of colonies originated from the water column of the Hármashegy pond. Several studies have shown that prolonged freezing (more than two years) without cryoprotectants of most *Microcystis* strains and their subsequent thawing did not lead to the death of all cells, and many of them restored their viability [38,39]. Thus, it is quite probable that in spring, a part of the toxigenic population of *Microcystis* from the ice remained viable, retained its toxigenicity and could inoculate the water column of Lake Ytyk-Kyuyol.

The maximum intracellular concentration of MC (803 ng L^−1^) was found in surface water sampled in August 2021. It was six times higher than the intracellular concentration (135 ng L^−1^) recorded in August 2022. However, the data were comparable with the level of the total MC content, including intracellular and extracellular fractions (400 ng L^−1^) detected in the water of Arctic lakes of southwestern Greenland over 2-year observations [18]. The recorded concentrations in the studied Lake Ytyk-Kyuyol, being a shallow eutrophic reservoir located in a continuous permafrost zone, were orders of magnitude lower than those in eutrophic freshwater ecosystems of temperate or tropical regions, which has been noted to be typical for northern reservoirs in severe climatic conditions [40]. However, it was reported that the MC concentrations in bloom spots in the circumpolar and Arctic lakes of Northwestern Russia could increase to 2500 ng L^−1^ in Lake Imandra of Kola Peninsula [16] and 12,500 ng L^−1^ in the Svyatozero Lake of the Onega basin [17]. Therefore, in the case of the appearance of blooming spots in the reservoir, the concentration of MCs can significantly increase, which indicates the need for further research. The recorded MC profile of arginine-containing congeners was common for the *Microcystis* spp. [41].

According to the registered MC concentrations, their level did not exceed the proposed guidelines, even for drinking water (1 µg L^−1^). However, due to the ability of MCs to accumulate in tissues, special attention should be paid to the use of lake water for irrigation and fishing in this lake. The increased anthropogenic load and global climate change may enhance the development of toxigenic cyanobacteria present in the lake, so further research is needed to assess the potential danger of cyanotoxins to humans.

The MC quota is the amount of cyanotoxin produced by the unit biomass of cyanobacteria producers (B) or by one cell. The MC quota is used as an indirect indicator of the toxicity of natural cyanobacterial blooms to assess potential risks to humans [42,43] and as an indicator of the intensity of MC production by cyanobacteria [44]. The calculated MC quotas for the studied lakes of the permafrost zone were extremely low and ranged from 0.2 × 10^−2^ to 0.31 μg MC mg^−1^ B during the study period. Such low values of MC quotas were noted in the polar Lake Imandra (0.5 × 10^−2^–6.9 × 10^−2^ μg MC mg^−1^ B) [16]. Compared to this, the MC quotas in reservoirs of the temperate zone reach 3.8 μg MC mg^−1^ B in the lakes of Germany [41] and up to 5.6 μg MC mg^−1^ B in reservoirs of the Volga, Kama and Don [45]. In our study, the calculated MC quotas significantly varied between seasons, with the maхimum values being 0.31 and 0.036 in 2021 and 2022, respectively. This fact was also noted in the polar lake [16]. Although the data are insufficient, it may be assumed that MC quotas are significantly lower in the northern water bodies compared to fresh waters of temperate latitudes. Further research is needed to elicit the reasons for low quotas in northern cyanobacteria populations and the lower MC production (expression of MC biosynthesis genes) under severe climatic conditions or a low proportion of producing strains. For example, it has recently been shown that only half of the analyzed colonies of *M. flos-aquae* contained the *mcy* genes for MC biosynthesis and are potentially capable of producing the toxin [46]. In our study, this species was among the dominant ones in the cyanobacterial community of the lake.

## 4. Conclusions

We found that in a small eutrophic lake located on the 62nd parallel north in the permafrost zone of Eurasia with the most severe climate in the Northern Hemisphere, MC-producing *Microcystis* were present year-round, both in the water under ice and frozen in ice. Even though *Microcystis* did not form under ice blooms, we assumed that its minor pelagic and ice populations could remain viable and be reintroduced into the water column in the spring. The calculated MC quotas for the studied lake were extremely low. Further research is needed for the elicitation of the reasons for low quotas in northern planktonic cyanobacteria populations.

## 5. Materials and Methods

### 5.1. Study Region and Lake Ytyk-Kyuyol

The studied water body is located in the zone of continuous permafrost on the 62nd parallel north. In winter, the Siberian anticyclone forming in the center of Asia and, in summer, frequent invasions of air masses from the Arctic Ocean with very low water vapor content strongly affect the weather conditions in the region. The climate is severely continental, with harsh, long winters and short, hot summers. The coldest inhabited place on Earth—Pole of Cold, Oymyakon—is situated in the region. According to www.worldclim.org, in the period of 1970–2000, the average annual temperature was 8.7 °С, with a minimum and maximum average monthly temperature of −41.4 °С and 25.0 °С, respectively. The length of an ice-free period in water bodies was 120–125 days [47].

The studied water body—Ytyk-Kyuyol Lake—is located on the terrace above the floodplain, in the middle reaches of the Lena River near Yakutsk, which is the largest city located in the permafrost zone (Figure 11). The population of Yakutsk is growing rapidly and has increased by almost 40% over the past 15 years, currently amounting to 330 thousand inhabitants. Lake Ytyk-Kyuyol is of a river origin. The waterbody is elongated, being 4400 m long and about 400 m wide, and the water surface area is 790.3 thousand m^2^. The maximum depth of the lake is 3 m. Ytyk-Kyuyol is divided by a dam equipped with a drainage pipe, providing a connection between the two parts of the lake. The lake is situated on the territory of the botanical garden. There are experimental sowing plots of the botanical garden, as well as summer cottages, on the shores of the lake. The lake is used for recreational purposes, for agricultural irrigation, as well as for household water supply.

### 5.2. Sampling

In total, 33 samples were collected in the period of 1 August 2021–25 November 2022 (Figure 1) from the surface layer (0–0.3 m), and the bottom layer at a 2 m depth. The coordinates of the sampling point were (62°01′22.0″ N 129°36′59.0″ E) (WGS 84). It was located on the northern side of the dam, within 30 m of the shore. The location of the sampling point on Lake Ytyk-Kyuyol was chosen based on the results of sighting at the planning stage of this study. During the vegetation season, we regularly noted a blooming spot at this place. The presence of such bloom spots is determined by a number of local factors related to the morphometry of the lake, the prevailing wind direction, etc. The choice of this location for one-year monitoring observations was necessary to obtain reliable information and achieve the aim of our study. Water samples were collected using the Wildco Water Sampler (Wildlife Supply Company, Yulee, FL, USA). Ice samples were taken of the whole ice cover thickness. The collected ice samples were melted at room temperature in the laboratory. The qualitative and quantitative planktonic samples were concentrated using an Apstein net (mesh size of 15 μm). The original volume of the quantitative planktonic sample was 30 L. The volume of the =concentrated planktonic sample was 15 mL. The conservation of all planktonic samples was performed by adding a 40% formalin solution (three drops per sample). Phytoplankton for the cyanotoxin analysis and the molecular genetic analysis was separated from natural water samples by pressurized filtration using 0.8 μm pore-size cellulose nitrate filters (Sartorius AG, Göttingen, Germany). The filters with precipitate were immediately frozen at −20 °C. Lake water (2 L) for hydrochemical analysis was transported to the laboratory for immediate analysis. The water and air temperature were measured with an electronic thermometer, Chektemp (Hanna Instruments, Woonsocket, RI, USA).

### 5.3. Water Chemistry Analysis

Chemical analyses were performed on water samples using standard methods [48,49]. The water color was determined using a photometric method. The pH was measured using a potentiometric method. The water salinity was calculated as the sum of ions using different methods: turbidimetry for sulphate anions, flame spectrophotometry for potassium and sodium cations, mercurimetry for chloride ions, and titration for calcium, magnesium and bicarbonate ions. The hardness of water was determined by complexometric titrations, using eriochrome black T as an indicator. A photometric method was applied to determine nutrient concentrations. Nessler’s reagent, Griess reagent and salicylic acid were used for the measurement of ammonium ion, nitrite ion and nitrate ion, respectively; ammonium molybdate was used for the measurement of phosphate ions. A combined reagent composed of ammonium molybdate, and ascorbic acid was used to determine the total phosphorus content. A photometric method was applied to determine the chemical oxygen demand (COD).

### 5.4. Algological Analysis

The qualitative and quantitative analyses of phytoplankton were performed using a light microscope at 550-fold magnification (Olympus BH-2, Olympus, Tokyo, Japan). Cell counting was carried out in a Nageotte chamber of 0.01 mL in triplicate. Biovolumes were determined as the cell count results multiplied by the means of the individual cell biovolume of one cell, according to [50]. Photomicrographs were obtained using a CK-13 digital camera (Lomo-microsystems, Saint Petersburg, Russia). The cyanobacterial species identification was performed according to [51,52,53]. The taxonomy of the species was obtained according to the data published on portal algabase.org.

### 5.5. Detection of Cyanotoxins via High-Performance Liquid Chromatography–Mass Spectrometry of a High Resolution

Cyanotoxin detection was performed in 3 groups of samples: surface water, bottom water and ice samples. Generally, intracellular MC concentrations were studied and the extracellular fraction was determined only in some summer samples of surface water.

To detect cyanotoxins, analytical-grade chemicals were used. Acetonitrile (HPLC-grade) and methanol (LiChrosolv hypergrade for LC–MS) were purchased from Merck (Darmstadt, Germany); formic acid (98–100%) was obtained from Fluka Chemika (Buchs, Switzerland). High-quality water (18.2 MΩ cm^−1^) was produced by the Millipore Direct-Q water purification system (Bedford, MA, USA). The MC-LR, MC-YR and MC-RR standards were purchased from Sigma Aldrich; MC-LA, MC-LF, MC-LY, MC-LW, [D-Asp^3^]MC-LR and [D-Asp^3^]MC-RR were from Enzo Life Sciences, Inc. New York, NY, USA.

To define the profile of toxins and quantify them, the high-performance liquid chromatography–high-resolution mass-spectrometry (HPLC–HRMS) method was used.

The sample preparation procedures were performed following [54], which involved the extraction of cyanotoxins from filtered water using SPE (Oasis HLB, Waters, Milford, MA, USA), and from the biomass separated from water sample by filtration using treatment with 75% methanol in an ultrasonic bath.

The analyses of extracts were performed using the LC-20 Prominence HPLC system (Shimadzu, Kyoto, Japan) coupled with a Hybrid Ion Trap-Orbitrap Mass Spectrometer—LTQ Orbitrap XL (Thermo Fisher Scientific, San Jose, CA, USA), according to [54]. Separation was achieved by gradient elution using a Thermo Hypersil Gold RP C18 column (100 × 3 mm, 3 μm) with the mobile phase consisting of water and acetonitrile, both containing 0.1% of formic acid.

Mass spectrometric analysis was carried out using electrospray ionization in the positive ion detection mode. The identification of target compounds was based on the accurate mass measurement of [М + Н]^+^ or [М + 2Н]^2+^ ions (resolution of 30,000, accuracy within 5 ppm), the collected fragmentation spectrum of the ions and the retention times. The concentrations of the toxins were calculated based on the peak area of the standards analyzed on the same day and under the same conditions.

MС quotas were calculated as the content of MCs per biomass of MC-producing *Microcystis* (determined using PCR).

### 5.6. Molecular Approach to Species Identification

For molecular species identification, DNA from environmental sample was extracted using the DNeasy Plant Mini Kit (Qiagen, Venlo, Limburg, The Netherlands). The amplification of the partial 16S rRNA gene and 16S–23S ITS rRNA region was performed using a pair of primers suggested by Wilmotte et al. [55] (1: 5′-CTC TGT GTG CCT AGG TAT CC-3′) and Neilan et al. [56] (27: 5′-AGA G TT TGA TCC TGG CTC AG-3′). PCR was carried out in 20 µL volumes using MasDDTaqMIX (Dialat Ltd., Moscow, Russia), 10 pmol of each oligonucleotide primer and 1 ng of DNA with the following amplification cycles: 3 min at 94 °C, 40 cycles (30 s at 94 °C, 40 s at 56 °C, 60 s at 72 °C) and a final extension time of 2 min at 72 °C. The amplified fragments were visualized on 1% agarose TAE gels via EthBr staining, purified using the QIAquick Gel Extraction Kit (Qiagen, Venlo, Limburg, The Netherlands), and then used as a template in sequencing reactions with the ABI PRISM^®^ BigDye™ Terminator v. 3.1 Sequencing Ready Reaction Kit, as per the standard protocol provided for the Applied Biosystems 3730 DNA Analyzer (Applied Biosystems, Waltham, MA, USA). Additionally, an internal primer (2: 5′-GGG GGA TTT TCC GCA ATG GG-3′) [57] was used in the sequencing procedure.

The sequence data for the tested sample were assembled using BioEdit 7.0.1 [58], and contained the 16S rRNA gene and 16S–23S ITS region. To find the most similar taxa, we performed a BLAST search with type material only. To test the phylogenetic affinity, we used the recently published database, CyanoSeq, of the cyanobacterial 16S rRNA gene [59]. The morphological species diversity in the genus *Microcystis* is not corresponded with the 16S rRNA sequence divergence, which placed some widely distributed morphologically accepted species into *Microcystis aeruginosa* based on less than 1% dissimilarity with strain type NIES-843 [31]. To perform the phylogenetic test, we additionally sampled 16S rRNA nucleotide sequences of nine strains assigned to five species. The total dataset contained the 16S rRNA gene (1432 cites) from 25 accessions, where *Chroococcopsis_gigantea*_SAG_12.99 was chosen as an outgroup.

The phylogeny was tested using the maximum likelihood (ML) method with IQ-TREE [60]. The ML analysis of the 16S rRNA gene included a search for the best-fit evolutionary model of nucleotide substitutions, with the incorporated option ModelFinder [61] and ultrafast bootstrapping [62] with 1000 replicates. The GTR + F+I model was selected as the best-fit evolutionary model with four rate categories of gamma distribution to evaluate the rate of heterogeneity among sites.

The similarity of the 16S rRNA gene region of the tested sample was calculated as the average pairwise *p*-distances in Mega 11 [63] using the pairwise deletion option for counting gaps with the following formula: 100 × (1 − p).

### 5.7. Molecular Approach to Test for Microcystin Biosynthesis Genes

Environmental DNA (eDNA) was extracted from plankton separated from the water and ice samples using the Diatom DNA Prep 200 reagent kit (Isogene Lab Ltd., Moscow, Russia), according to the manufacturer’s instructions.

eDNA was used to search for the *mcyA* and *mcyE* genes involved in the synthesis of MC. The general *mcy*ACd [64], HEP [65] and *Microcystis*-specific mcyA_MF/MR [66], *mcy*EF2/Mic*mcy*E-R8 [67] primers were used for the amplification of *mcy* genes. All PCR reactions were carried out at a volume of 25 μL by using the DreamTaq PCR Master Mix (Thermo Scientific, Waltham, MA, USA). The amplification program was as follows: initial DNA denaturation for 3 min at 95 °C; 37 amplification cycles (one cycle was 30 s at 95 °C, 30 s at 58 °C and 1 min at 72 °C); and a final elongation for 10 min at 72 °C. The PCR products were separated by electrophoresis on a 1.5% agarose gel, stained with ethidium bromideand visualized under UV illumination. The size of the amplified DNA fragments was determined using the 100–1000 bp molecular weight marker (Isogene Lab. Ltd., Moscow, Russia).

### 5.8. PCR Products of mcyA and mcyE Sequencing and Phylogenetic Analysis

To verify that the *mcyA* and *mcyE* genes were amplified, some PCR products were sequenced using an ABI PRISM BigDye Terminator cycle sequencing kit (version 3.1) in a 3730 DNA Analyzer (Applied Biosystems, Waltham, MA, USA), at the Center for Collective Use “Genome” (Moscow, Russia). The sequences obtained for both strands were assembled using BioEdit v7.1.3 [68] and analyzed with BLAST to find appropriate similarities for phylogenetic estimation. The obtained DNA sequences were aligned with other sequences of *mcyA* and *mcyE* genes from the GenBank using the Muscle algorithm [69]. The evolutionary distances were computed using the Tamura 3-parameter method, and the rate variation among sites was modeled with a gamma distribution. Phylogenetic trees were constructed via neighbor-joining (NJ) algorithms using MEGA 6.0 software [70]. The node support was estimated with bootstrap analysis (1000 replicates).

### 5.9. Statistical Analysis

Non-parametric methods of biostatistics were used. The Spearman correlation coefficient (r_s_) was used to identify the relationships between *Microcystis* cell density and cell-bound MC concentrations. The testing of the mean difference between paired and unpaired samples was carried out using the Wilcoxon matched pairs test and the Mann–Whitney U test, respectively. The critical significance level in this study was taken as *р* = 0.05. Statistical analysis of the data was performed with PAST 4.04 [71].

## Figures and Tables

**Figure 1 toxins-15-00467-f001:**
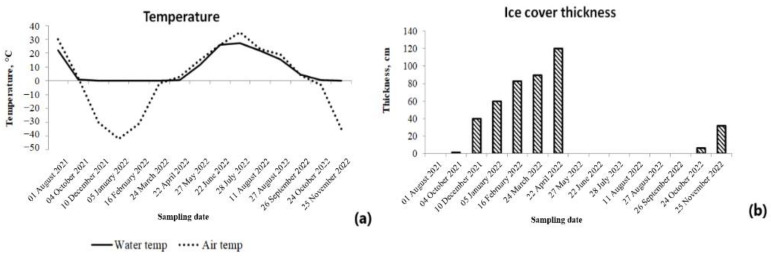
Some physical parameters measured in the Ytyk-Kyuyol Lake during sampling: (**a**) Water and air temperature; (**b**) Ice cover thickness.

**Figure 2 toxins-15-00467-f002:**
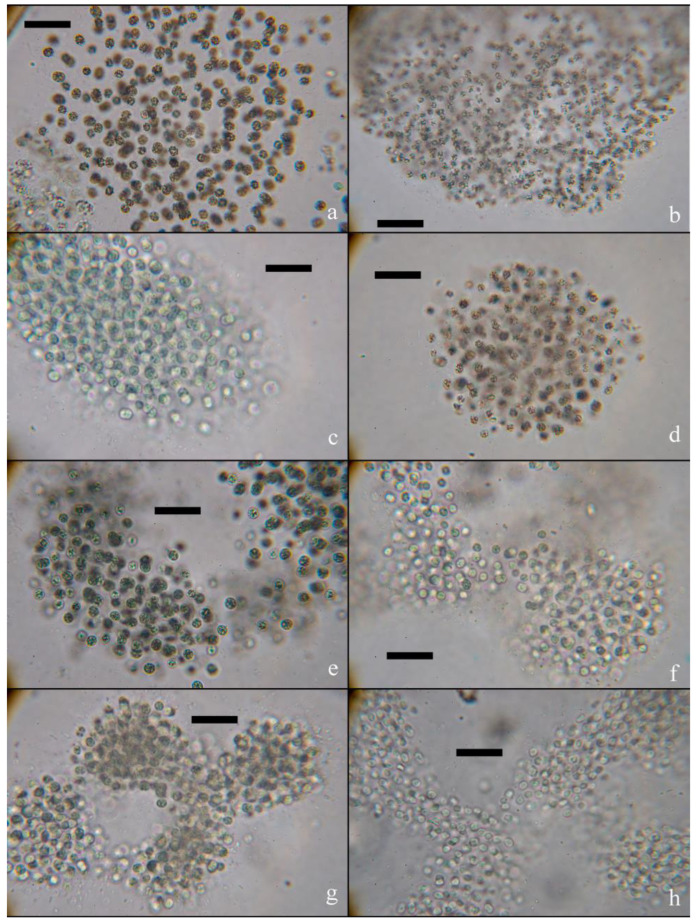
Colonies of *M. flos-aquae*: (**a**) in water in July, (**b**) under-ice water in November, (**c**) ice in November and (**d**) ice in April, and (**e**) colonies of *M. aeruginosa* in water in July, (**f**) under-ice water in November, (**g**) ice in November and (**h**) ice in April. Scale bar—20 μm.

**Figure 3 toxins-15-00467-f003:**
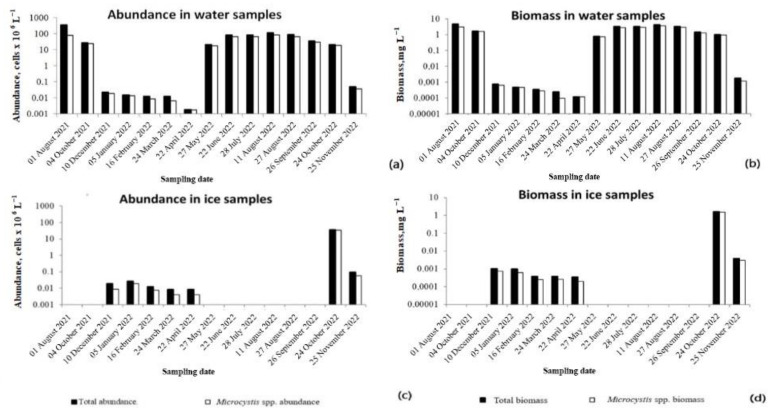
Abundance and biomass of cyanobacteria and potentially toxigenic *Microcystis* spp. (*M. aeruginosa*, *M. botrys*, *M. flos-aquae*, *M. wesenbergii*, *M. novacekii)*: (**a**,**b**) in the surface layer of water and (**c**,**d**) ice samples, respectively. Histograms are plotted on a logarithmic scale. The calculated abundance (Wilcoxon matched pairs test, T = 26, *p* = 0.53) and biomass (Wilcoxon matched pairs test, T = 22, *p* = 0.33) of cyanobacteria in the samples collected both on the surface and at the bottom differed statistically non-significantly. In this regard, only data from the surface water layer are presented in the figure. All raw data for surface and bottom water samples are given in (Appendix A). The standard deviation for the abundance and biomass of the studied samples are given in Appendix A.

**Figure 4 toxins-15-00467-f004:**
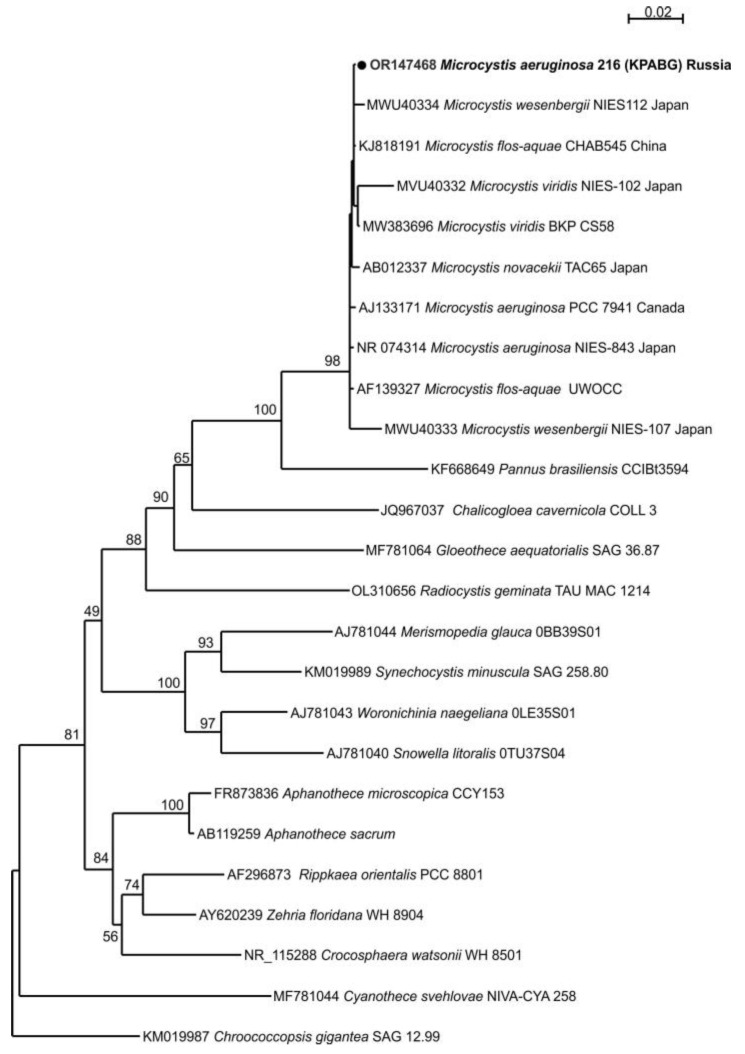
Phylogram obtained under the maximum likelihood approach for 25 accessions of nearest relatives to the genus *Microcystis* based on the 16S rRNA gene. Bootstrap support values greater than 50% are indicated, and GenBank accession numbers are provided.

**Figure 5 toxins-15-00467-f005:**
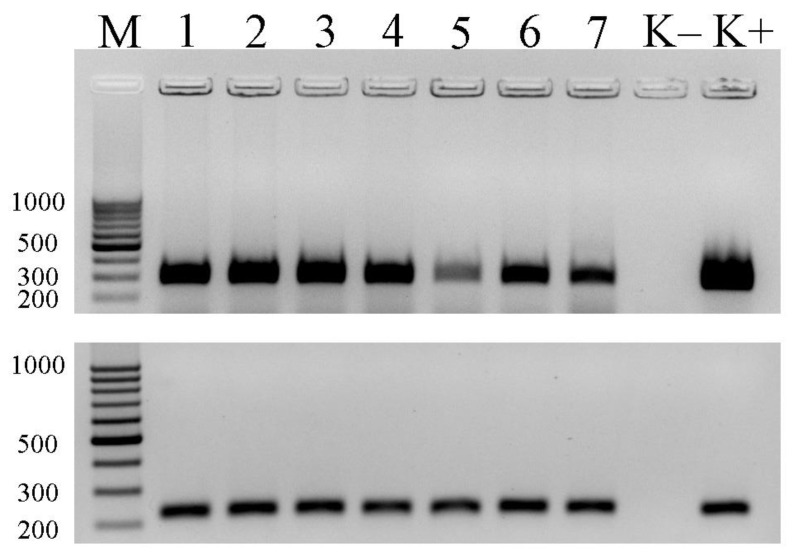
Gel electrophoresis of the PCR products for *mcyA* gene using: (**top**) a general mcyACd primer set and (**bottom**) *Microcystis*-specific *mcy*A_MF/MR primer set. M—100 bp molecular marker. Lаnes: 1—ice sample (10 December 2021), 2—ice sample (5 January 2022), 3—ice sample (16 February 2022), 4—ice sample (24 March 2022), 5—ice sample (22 April 2022), 6—ice sample (24 October 2022), 7—ice sample (25 November 2022), (K−)—negative control (no template), (K+)—positive control (DNA of *Microcystis aeruginosa* PCC 7806).

**Figure 6 toxins-15-00467-f006:**
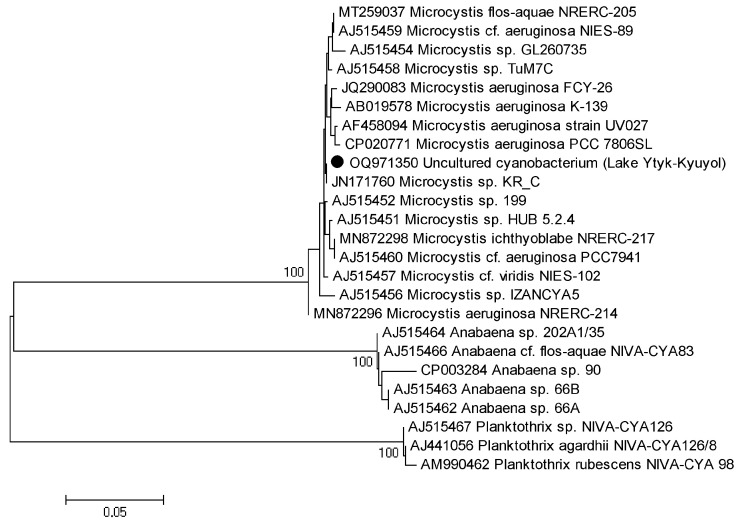
Neighbor-joining unrooted phylogenetic tree based on *mcyA* sequences. Genbank accession numbers of sequences are shown on the tree. Bootstrap values (>70%) are given at the nodes. The scale bar represents 5 nucleotide substitutions per 100 nucleotides.

**Figure 7 toxins-15-00467-f007:**
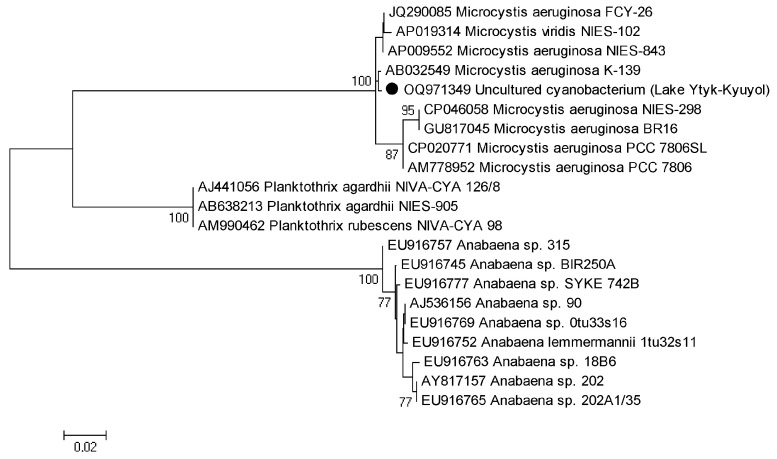
Neighbor-joining unrooted phylogenetic tree based on *mcyE* sequences. Genbank accession numbers of the sequences are shown on the tree. Bootstrap values (>70%) are given at the nodes. The scale bar represents 2 nucleotide substitutions per 100 nucleotides.

**Figure 8 toxins-15-00467-f008:**
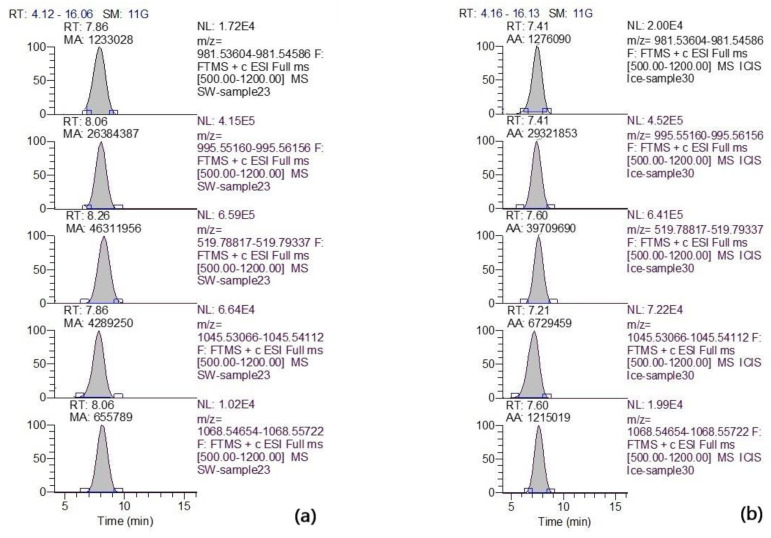
Extracted high-resolution ion chromatogram (mass accuracy within 5 ppm) for MC congeners (from top to bottom): m/z 981.54095 ([M + H]^+^ [D-Asp^3^]MC-LR); m/z 995.55658 ([M + H]^+^ MC-LR); m/z 519.79077 ([M + 2H]^2+^ MC-RR); m/z 1045.53589 ([M + H]^+^ MC-YR); m/z 1068.55188 ([M + H]^+^ MC-WR) detected in the biomass filtered (**a**) from surface water (sample volume 0.5 L) and (**b**) ice (sample volume 1.0 L).

**Figure 9 toxins-15-00467-f009:**
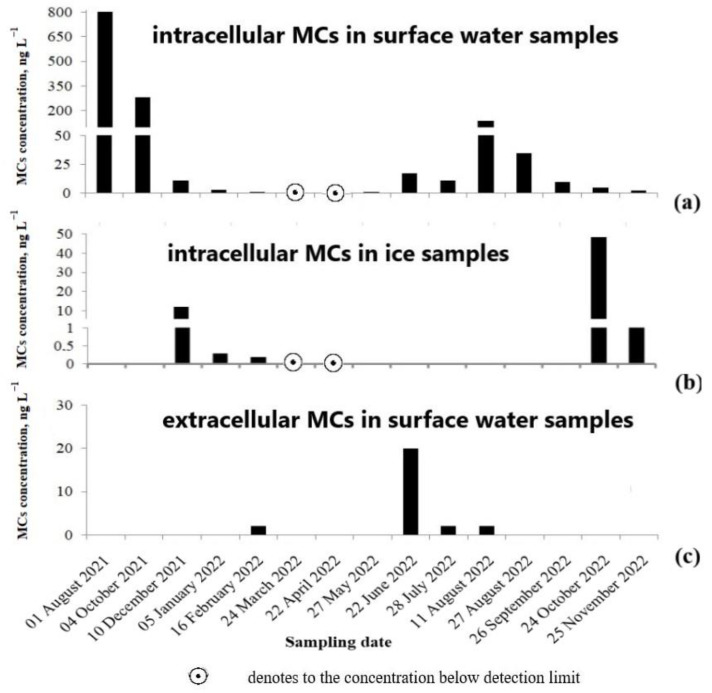
Microcystin concentration in the samples of water and ice cover from the studied Lake Ytyk-Kyuyol: (**a**) intracellular MC concentration in surface water samples, (**b**) intracellular MC concentration in ice samples, and (**c**) extracellular MC concentration in surface water samples. Since no statistically significant differences were found between the concentrations of MCs measured in surface and bottom water samples (Wilcoxon matched pairs test, T = 16, *p* = 0.78), only data for the surface water layer are presented in the figure. All raw data obtained for the surface and bottom water samples are given in Appendix A.

**Figure 10 toxins-15-00467-f010:**
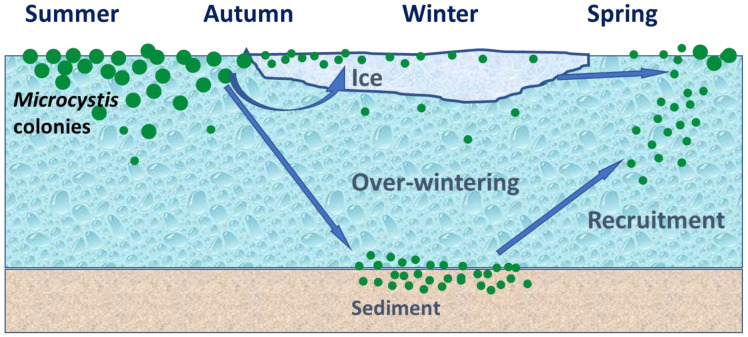
Scheme of the possible strategy followed by microcystin-containing *Microcystis* colonies to survive under the conditions of dramatically low temperatures and long ice cover periods in the permafrost zone.

**Figure 11 toxins-15-00467-f011:**
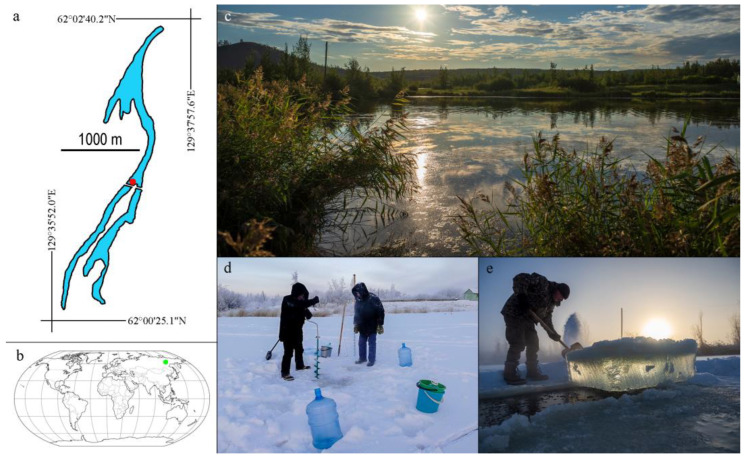
The study area. Map of the Ytyk-Kyuyol Lake with a red dot indicating the sampling point (**a**) and World map with a green point showing the geographic location of the Ytyk-Kyuyol Lake (**b**). Sampling station. Lake Ytyk-Kyuyol in August 2022 (**c**); ice drilling during sampling in January 2022 (**d**); ice sample collecting in November 2022 (**e**).

**Table 1 toxins-15-00467-t001:** Chemical variables of the Ytyk-Kyuyol Lake water, measured during specific sampling dates in 2022.

Variable	5 January	22 April	27 May	28 July	11 August	26 September
pH	7.45	7.38	9.13	8.74	9.23	7.41
Salinity, mg L^−1^	509.7	704.4	276.4	475.0	349.5	461.8
Hardness, mmol L^−1^	4.0	6.2	2.7	4.0	3.8	4.3
N–NH_4_, mg L^−1^	0.95	0.49	0.28	0.32	0.62	0.14
N–NO_3_, mg L^−1^	7.61	6.50	0.84	0.88	2.3	0.75
N–NO_2_, mg L^−1^	0.75	0.41	0.02	0.03	0.11	0.02
P tot, mg L^−1^	0.24	0.63	0.23	0.46	0.18	0.58
PО_4_, mg L^−1^	0.06	0.27	0.06	0.03	0.04	0.04
Color, Pt/Co grad.	62.0	83.0	38.0	87.0	119.0	45.0
COD, mg O L^−1^	82.4	79.8	46.8	47.2	84.5	42.0

**Table 2 toxins-15-00467-t002:** The value of nucleotide sequence similarity for *Microcystis* strains based on nucleotide sequence data of the 16S rRNA gene, %.

	Strain	1	2	3	4	5	6	7	8	9
1	OR147468 *Microcystis aeruginosa* 216 Russia									
2	MWU40334 *M. wesenbergii* NIES112 Japan	99.57								
3	KJ818191 *M. flos-aquae* CHAB545 China	99.86	99.57							
4	MVU40332 *M. viridis* NIES-102 Japan	98.56	98.28	98.56						
5	MW383696 *M. viridis* BKP CS58	99.71	99.43	99.71	98.71					
6	AB012337 *M. novacekii* TAC65 Japan	99.57	99.43	99.57	98.42	99.57				
7	AJ133171 *M. aeruginosa* PCC7941	99.57	99.42	99.57	98.27	99.42	99.57			
8	NR 074314 *M. aeruginosa* NIES-843 Japan	99.64	99.35	99.64	98.49	99.64	99.50	99.64		
9	AF139327 *M. flos-aquae* UWOCC	99.64	99.35	99.64	98.35	99.50	99.64	99.64	99.71	
10	MWU40333 *M. wesenbergii* NIES-107 Japan	98.71	98.42	98.71	97.55	98.70	98.71	98.70	98.78	98.78

## Data Availability

The data presented in this study are available in Appendix A.

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
