# Peer review of "Year-Round Presence of Microcystins and Toxin-Producing Microcystis in the Water Column and Ice Cover of a Eutrophic Lake Located in the Continuous Permafrost Zone (Yakutia, Russia)"

_toxins, 2023, doi:10.3390/toxins15070467_

Round 1

Reviewer 1 Report

Dear authors, excellent work! I have a few remarks on your paper: 

Lines 111-113 - Please use Latin for the species names. 

Line 124 - I highly recommend you include the species M. wesenbegii as a potential MC producer.

Please see the following reference: Microcystis aeruginosa and M. wesenbergii Were the Primary Planktonic Microcystin Producers in Several Bulgarian Waterbodies (August 2019). Appl. Sci. 2021, 11, 357. https://doi.org/10.3390/app11010357.

Line 144 - There is a discrepancy between this conclusion and the conclusion from lines 108 and 113. Have you detected both species or M. flos - aquae only? 

Line 339 - Please specify the breviation "B - is this biomass? In the cited reference the values are presented in μg/mg

Author Response

Reviewer 1

Dear Reviewer,

Thank you for your positive evaluation of our study and your valuable comments.

The corrections made are listed below.

Lines 111-113 - Please use Latin for the species names. 

Answer: Corrected.

Line 124 - I highly recommend you include the species M. wesenbegii as a potential MC producer.

Please see the following reference: Microcystis aeruginosa and M. wesenbergii Were the Primary Planktonic Microcystin Producers in Several Bulgarian Waterbodies (August 2019). Appl. Sci. 2021, 11, 357. https://doi.org/10.3390/app11010357.

Answer: We included M. wesenbergii into potential MC producers.

Line 144 - There is a discrepancy between this conclusion and the conclusion from lines 108 and 113. Have you detected both species or M. flos - aquae only? 

Answer:

Changes to the sentences in lines 108 and 113 have been made to avoid misunderstanding.  “…by the end of the period of ice phenomena, only one species (Microcystis flos-aquae) was found in the under-ice plankton sample collected in April…” and  ” M. flos-aquae was the species found in all examined ice samples”, respectively.

Line 144 The sentence was deleted. Thank you for your remark.

Line 339 - Please specify the breviation "B - is this biomass? In the cited reference the values are presented in μg/mg.

Answer: We specified the breviation “B” in the following sentence: “The MC quota is the amount of cyanotoxin produced by a unit biomass of cyanobacteria producers (B) or by one cell”.

Reviewer 2 Report

Year-round presence of microcystins and toxin-producing Microcystis in water column and ice cover of a eutrophic lake located in the continuous permafrost zone (Yakutia, Russia)

This is an interesting and well-presented manuscript. There are a few small details that should be revised before publication.

38: Replace Last years by In recent years

50 winter, under ice,

Does the exposure to freezing change the toxicity of the cyanobacteria

Line 110-114-Italics

Indicate what mln stands for in Fig. 3

144 Replace To by At

159 Replace - Obtained results allowed us to consider  tested strain as Microcystis aeruginosa by

Our results indicate that the tested strain was Microcystis aeruginosa

Rewrite lines 214-215; the meaning of the sentence is not clear

226-What does the B at the end of the sentence refer to?

234-Delete of vegetation

241 the shallow

245 Replace permanently presented with were permanently present

265-266-The statement should be rephrased; it is not clear. Why is depositary highlighted?

Could the decrease in microcystins mentioned in lines 295-296 be related to the low predation pressure under permafrost conditions?

Line 342 maxmall?

Based on the uses mentioned in the Methods and the concentration of cyanotoxins measured, the authors should add a paragraph, in the Discussion, on the applications of their findings to the use of water from this pond by the local community.

Minor changes

Author Response

Reviewer 2

Dear Reviewer,

Thank you for your positive evaluation of our study and your valuable comments.

The corrections made are listed below.

Line 38: Replace Last years by In recent years

Answer: Corrected

Line 50: winter, under ice,

Answer: Corrected

Does the exposure to freezing change the toxicity of the cyanobacteria

Answer: We are not aware of published studies showing that freezing of Microcystis cells affects their toxigenicity. This is a very interesting, but apparently unexplored question, which we cannot answer based on the design of our study. To answer this question, it is necessary to set up model laboratory experiments on freezing/thawing of Microcystis strains, followed by analysis of the level of MC production.

Line 110-114-Italics

Answer: Corrected

Indicate what mln stands for in Fig. 3

Answer: We changed “mln cells L-1” to “cells x 106 L-1

Line 144 Replace To by At 

Answer: Corrected

Line 159 Replace - Obtained results allowed us to consider tested strain as Microcystis aeruginosa by

Our results indicate that the tested strain was Microcystis aeruginosa

Answer: Replaced

Rewrite lines 214-215; the meaning of the sentence is not clear

Answer: The sentence was rewritten to “Throughout the year, only two out of 15 surface water samples (sampling dates – 24 March and 22 April) did not contain intracellular MCs (detection limit, LOD 0.1 ng L−1)”.  We meant that surface water always contained MCs except two samples.

Line 226 What does the B at the end of the sentence refer to?

Answer: “B’  is a unit biomass of cyanobacteria producers. We added the explanation to the text.

Line 234 Delete of vegetation

Answer: deleted

Line  241 the shallow

Answer: An article was added.

Line 245 Replace permanently presented with were permanently present

Answer: corrected

Line 265-266 The statement should be rephrased; it is not clear. Why is depositary highlighted?

Answer: The sentence was rephrased: “It has been known that lake sediments is the main depositary for the over-wintering population of Microcystis keeping the viability”. We meant that the main part of living cells is overwintering in the sediments.

Could the decrease in microcystins mentioned in lines 295-296 be related to the low predation pressure under permafrost conditions?

Answer: In these lines, we discussed the concentrations of intracellular microcystins and toxigenic Microcystis cells frozen into ice. We suppose that predators could not influence on the intracellular microcystins concentration in ice.

Line 342 maxmall?

Answer: Corrected

Based on the uses mentioned in the Methods and the concentration of cyanotoxins measured, the authors should add a paragraph, in the Discussion, on the applications of their findings to the use of water from this pond by the local community.

Answer: The paragraph was added in the discussion section:

“According to the registered MC concentrations, their level did not exceed the proposed guidelines even for drinking water (1 µg l-1). However, due to the ability of MC to accumulate in tissues, special attention should be paid to the use of lake water for irrigation and fishing in this lake. The increased anthropogenic load and global climate change may enhance the development of toxigenic cyanobacteria present in the lake, so further research is needed to assess the potential danger of cyanotoxins to humans”.

Reviewer 3 Report

The manuscript titled “Year-round of microcystins and toxin-producing Microcystis in water colum and ice cover of a eutrophic lake located in the continuous permafrost zone (Yakutia, Rusia)” by anonimous authors is an original work where the viability of Microcystis cyanobacteria  and the toxic produced microcystins is tested in ultrafrozen water reservoirs. The authors demonstrated that Microcystins are capable to survive under permafrost conditions. Moreover, the authors provide an hypothesis to explain the observations found in this work. The study is interesting and it is well-designed.

However, it exists some points that need to be addressed (please, see them below detailed point-by-point). The most relevant outcomes found by the authors can contribute to better understand the underlying survival mechanisms of Microcystins. For this reason, I will recommend the present scientific manuscript for further publication in Toxins once all the below described suggestions will be properly fixed.

Here, there exists some points that must be covered in order to improve the scientific quality of the manuscript paper:

1) ABSTRACT. “Cyanobacterial biomass ranged from 0.0001 to 4.8 mg L-1, (…)” (lines 9-10). Please, the authors modify the aforementioned statement by “(…) from 1.0 10-4 to 4.8 mg L-1” in order to homogenize the significant figures. This comment should be taken into account for the rest of the main manuscript body text (e.g. “The calculated MCs quotas were in range of 0.002-0.31 µg MC mg-1”, line 226).

2) INTRODUCTION. “Cyanobacteria colonize a wide range (…). One of the harmful features (…) to produce a wide range of (…)” (lines 30-32). Please, the authors need to carefully check out the English in order to avoid repetitions or typos in the manuscript paper. Please, the authors should consider to change one of the “wide range of” terms by “broad panoply of”.

3) The authors should discuss in the introduction section the most recent advances in the use of ultrasensitive techniques like biomonitoring [1] or single molecule [2] tools to detect the citotoxicity of microcystins [3,4], respectively.

[1] Zhou, Q.; et al. Biomonitoring an appeling tool for assessment of metal pollution in the aquatic ecosystem. Anal. Chim. Acta 2008, 606, 135-150. https://doi.org/10.1016/j.aca.2007.11.018.

[2] Marcuello, C. Present and future opportunities in the use of atomic force microscopy to address the physico-chemical properties of aquatic ecosystems at the nanoscale level. Int. Aquat. Res. 2022, 14, 231-240. https://doi.org/10.22034/IAR.2022.1965012.1317.

[3] Laughinghouse 4th, H.D.; et al. Biomonitoring genotoxicity and cytotoxicity of Microcystis aeruginosa (Chroococcales, cyanobacteria) using the Allium cepta test. Sci. Total Environ. 2012, 432, 180-188. https://doi.org/10.1016/j.scitotenv.2012.05.093.

[4] Ceballos-Laita, L.; et al. Microcystin-LR Binds Iron, and Iron Promotes Self-Assembly. Environ. Sci. Technol. 2017, 51, 4181-4850. https://doi.org/10.1021/acs.est.6b05939.

4) “(…) only one report of a natural toxigenic population of Microcystis whose cells remained viable after freezing (…)” (lines 59-61). Here, even if I fully agree with this statement that highlights the grade of novelty of this work, it may be convenient to point out the impact of environmental changes on the viability and survival rate of cyanobacterial populations [5].

[5] Rodríguez Tito, J.C.; et al. First Report on Microcystin-LR Ocurrence in Water Reservoirs of Eastern Cuba, and Environmental Trigger Factors. Toxins 2022, 14, 209. https://doi.org/10.3390/toxins14030209.

5) RESULTS. “Waters are characterized by elevated COD values” (line 91). Please, the authors should define the term “chemical oxygen demand”. Then, the abbreviation “COD” should be placed between brackets.

6) Figure 2 (line 116). Please, the authors should indicate the scale bar associated to the lateral dimensions of 20 µm of the optical microscope images (panels a-h).

7) Figure 3 (line 136). Why did the authors not plot the standard deviation (SD) values of the measured parameteres? The authors specified in the respective M&M section (lines 394-395) the number of samples (33) collected in the present study. Same comment for the Figure 9 (line 227).

8) (OPTIONAL) The authors already conducted the Wilcoxon matched pair test for the calculated abundance and cyanobacteria biomass. Nevertheless, this information should be pointed out in the respective Figure 7 (line 136) and Figure 9 (line 228) captions.

9) DISCUSSION. “Microcytis colonies due to bouyancy control can sink to the bottom (….) summer period” (lines 267-274). Here, the authors provide a hypothesis about the strategy followed by Microcystins to survive under dramatically low temperature conditions. It may be desirable to add a schematic representation to illustrate the potential readers abouth this feasible hypothesis proposed by the authors. Then, please, the authors should change the term “Microcytis” by “Microcystins”.

10) CONCLUSIONS. This section is clear and the most outcomes found by the authors are clearly shown. No actions are requested.

11) MATERIALS AND METHODS. “Study region and Lake Ytyk-Kyuyol” (lines 364-392). Here, it is the most critical point and my main concern about this work. The authors should perfectly indicate the reasons to acquire the samples only in one single point of the Ytyk-Kyuyol lake. Could this point affect to the credibility of the gathered data? (The authors already pointed out some insights during the Conclusions section in this regard but, it may be convenient to provide some brief further discussion for this point).

12) REFERENCES. The references are mostly in the proper format style of Toxins. Nevertheless, the authors should take care about some of them (e.g. reference number 32 where the jorunal name is not in abbreviated form).

Moderate editing of English language required. The authors should check the English before to submit the revised manuscript version.

Author Response

Reviewer 3

Dear Reviewer,

Thank you for your positive evaluation of our study and your valuable comments.

The corrections made are listed below.

Here, there exists some points that must be covered in order to improve the scientific quality of the manuscript paper:

1)ABSTRACT. “Cyanobacterial biomass ranged from 0.0001 to 4.8 mg L-1, (…)” (lines 9-10). Please, the authors modify the aforementioned statement by “(…) from 1.0 10-4 to 4.8 mg L-1” in order to homogenize the significant figures. This comment should be taken into account for the rest of the main manuscript body text (e.g. “The calculated MCs quotas were in range of 0.002-0.31 µg MC mg-1”, line 226).

Answer: Corrected

2) INTRODUCTION. “Cyanobacteria colonize a wide range (…). One of the harmful features (…) to produce a wide range of (…)” (lines 30-32). Please, the authors need to carefully check out the English in order to avoid repetitions or typos in the manuscript paper. Please, the authors should consider to change one of the “wide range of” terms by “broad panoply of”.

Answer: Thank you for suggestion, I changed the second sentence.

3) The authors should discuss in the introduction section the most recent advances in the use of ultrasensitive techniques like biomonitoring [1] or single molecule [2] tools to detect the citotoxicity of microcystins [3,4], respectively.

[1] Zhou, Q.; et al. Biomonitoring an appeling tool for assessment of metal pollution in the aquatic ecosystem. Anal. Chim. Acta 2008606, 135-150. https://doi.org/10.1016/j.aca.2007.11.018.

[2] Marcuello, C. Present and future opportunities in the use of atomic force microscopy to address the physico-chemical properties of aquatic ecosystems at the nanoscale level. Int. Aquat. Res202214, 231-240. https://doi.org/10.22034/IAR.2022.1965012.1317.

[3] Laughinghouse 4th, H.D.; et al. Biomonitoring genotoxicity and cytotoxicity of Microcystis aeruginosa (Chroococcales, cyanobacteria) using the Allium cepta test. Sci. Total Environ2012432, 180-188. https://doi.org/10.1016/j.scitotenv.2012.05.093.

[4] Ceballos-Laita, L.; et al. Microcystin-LR Binds Iron, and Iron Promotes Self-Assembly. Environ. Sci. Technol201751, 4181-4850. https://doi.org/10.1021/acs.est.6b05939.

Answer: Thank you for your suggestion. We have inserted an additional sentence in Introduction section after the aim of the work, explaining our choice of three methodological approaches in our work from the whole set of methods.

4) “(…) only one report of a natural toxigenic population of Microcystis whose cells remained viable after freezing (…)” (lines 59-61). Here, even if I fully agree with this statement that highlights the grade of novelty of this work, it may be convenient to point out the impact of environmental changes on the viability and survival rate of cyanobacterial populations [5].

[5] Rodríguez Tito, J.C.; et al. First Report on Microcystin-LR Ocurrence in Water Reservoirs of Eastern Cuba, and Environmental Trigger Factors. Toxins 202214, 209. https://doi.org/10.3390/toxins14030209.

Answer: The first paragraph in the Introduction is devoted to a brief discussion of environmental factors affecting cyanobacteria populations (eutrophication, climate change, high and low temperatures). In this part of the Introduction, we have inserted the suggested reference to the article by Tito et al., 2022

5) RESULTS. “Waters are characterized by elevated COD values” (line 91). Please, the authors should define the term “chemical oxygen demand”. Then, the abbreviation “COD” should be placed between brackets.

Answer: corrected

6) Figure 2 (line 116). Please, the authors should indicate the scale bar associated to the lateral dimensions of 20 µm of the optical microscope images (panels a-h).

Answer: Scale bars were modified to clear perception. The dimension of scale bar (20 μm) is mentioned in the caption to the Figure 2.

7) Figure 3 (line 136). Why did the authors not plot the standard deviation (SD) values of the measured parameteres? The authors specified in the respective M&M section (lines 394-395) the number of samples (33) collected in the present study. Same comment for the Figure 9 (line 227).

Answer: Figure 3 is plotted in logarithmic scale, plotted sd values will not be visible on the histogram. Therefore, we added an additional table to the Supplementary Materials, where we provided data on the standard deviation for the average values of abundance and biomass for each sample."

Comments for figure 9 (line 227):

Mass spectrometric analysis was performed only once for each of 33 samples. The idea of the study was to assess the temporal changes in the concentration in the lake. Therefore, I can’t calculate the standard deviation of the determined value for each sample.

8) (OPTIONAL) The authors already conducted the Wilcoxon matched pair test for the calculated abundance and cyanobacteria biomass. Nevertheless, this information should be pointed out in the respective Figure 7 (line 136) and Figure 9 (line 228) captions.

Answer: We moved the paragraphs containing this information from the section Materials and Methods to the captions for Figure 3 and Figure 9

9) DISCUSSION. “Microcytis colonies due to bouyancy control can sink to the bottom (….) summer period” (lines 267-274). Here, the authors provide a hypothesis about the strategy followed by Microcystins to survive under dramatically low temperature conditions. It may be desirable to add a schematic representation to illustrate the potential readers abouth this feasible hypothesis proposed by the authors. Then, please, the authors should change the term “Microcytis” by “Microcystins”.

Answer: Thank you for the interesting proposal to add a schematic representation our feasible hypothesis. The Figure containing the scheme was added to the text. Microcytis was corrected to Microcystis.

Figure Scheme of possible strategy followed by microcystin-containing Microcystis colonies to survive under conditions of dramatically low temperature and long ice-cover period in the permafrost zone

10) CONCLUSIONS. This section is clear and the most outcomes found by the authors are clearly shown. No actions are requested.

11) MATERIALS AND METHODS. “Study region and Lake Ytyk-Kyuyol” (lines 364-392). Here, it is the most critical point and my main concern about this work. The authors should perfectly indicate the reasons to acquire the samples only in one single point of the Ytyk-Kyuyol lake. Could this point affect to the credibility of the gathered data? (The authors already pointed out some insights during the Conclusions section in this regard but, it may be convenient to provide some brief further discussion for this point).

Answer: The location of the sampling point on Lake Ytyk-Kyuyol was chosen based on the results of visual observations at the planning stage of this study. During the warm season, we have regularly noted a blooming spot at this place. The presence of such bloom spots is determined by a number of local factors related to the morphometry of the lake, the prevailing wind direction, etc. The choice of this location for one-year monitoring observations was necessary to obtain reliable information and achieve the goal set in the work. 

12) REFERENCES. The references are mostly in the proper format style of Toxins. Nevertheless, the authors should take care about some of them (e.g. reference number 32 where the jorunal name is not in abbreviated form).

Answer: Thank you for the remark. In references 32 и 52, the journal title corrected to the abbreviated form “ Appl Environ Microbiol

Round 2

Reviewer 3 Report

The authors have done a great effort in order to significantly improve the scientific quality of the manuscript. For this reason and according the originality of the conducted research and the scope of the journal, I warmly recommend this work for further publication in Toxins.